# Tracking Animal Reservoirs of Pathogenic *Leptospira*: The Right Test for the Right Claim

**DOI:** 10.3390/tropicalmed6040205

**Published:** 2021-11-30

**Authors:** Yann Gomard, Koussay Dellagi, Steven M. Goodman, Patrick Mavingui, Pablo Tortosa

**Affiliations:** 1Unité Mixte de Recherche Processus Infectieux en Milieu Insulaire Tropical (UMR PIMIT), Université de La Réunion, CNRS 9192, INSERM U1187, IRD 249, Plateforme Technologique CYROI, 2 Rue Maxime Rivière, 97490 Sainte-Clotilde, France; yann.gomard@gmail.com (Y.G.); patrick.mavingui@cnrs.fr (P.M.); 2International Department, Institut Pasteur, 75015 Paris, France; koussay.dellagi@pasteur.fr; 3Field Museum of Natural History, 1400 South Lake Shore Drive, Chicago, IL 60605, USA; sgoodman@fieldmuseum.org; 4Association Vahatra, BP 3972, Antananarivo 101, Madagascar

**Keywords:** *Leptospira*, animal reservoir, One Health, MAT

## Abstract

Leptospirosis is the most prevalent bacterial zoonosis worldwide and, in this context, has been extensively investigated through the One Health framework. Diagnosis of human leptospirosis includes molecular and serological tools, with the serological Microscopic Agglutination Test (MAT) still being considered as the gold standard. Mammals acting as reservoirs of the pathogen include species or populations that are able to maintain chronic infection and shed the bacteria via their urine into the environment. Animals infected by *Leptospira* are often identified using the same diagnosis tool as in humans, i.e., serological MAT. However, this tool may lead to misinterpretations as it can signal previous infection but does not provide accurate information regarding the capacity of animals to maintain chronic infection and, hence, participate in the transmission cycle. We employ in this paper previously published data and present original results on introduced and endemic small mammals from Indian Ocean islands to show that MAT should not be used for the identification of *Leptospira* reservoirs. By contrast, serological data are informative on the level of exposure of animals living in a specific environment. We present a sequential methodology to investigate human leptospirosis in the One Health framework that associates molecular detection in humans and animals, together with MAT of human samples using *Leptospira* isolates obtained from reservoir animals occurring in the same environment.

## 1. Introduction

Leptospirosis is claimed as the most widespread bacterial zoonosis worldwide causing over 1 million human cases and nearly 60,000 deaths annually [1]. Despite its medical and veterinary importance, the burden of the disease remains underestimated in numerous countries, stimulating epidemiological investigations conducted in the One Health framework and aiming at identifying the major drivers of the disease [2,3,4]. Leptospirosis is caused by pathogenic bacteria belonging to the genus *Leptospira* (family Leptospiraceae), which have been historically classified using antigenic determinants through the Microscopic Agglutination Test (MAT) [5] and Co-Agglutination Absorption Technique (CAAT), allowing researchers to define over 20 serogroups and 300 serovars [6,7]. Molecular tools have been more recently developed and have revealed a higher genetic diversity of *Leptospira* than previously recognized [7,8,9] with several additional species being discovered using genomic approaches [10,11].

The main biological cycle of pathogenic *Leptospira* involves wild or domestic animals acting as reservoirs that shed the bacteria via their urine into the environment [12]. Humans get mostly (but not only, see [13]) infected through indirect contact with a contaminated environment. Although virtually all mammal species can get infected by pathogenic *Leptopsira*, some requirements are needed to consider them as reservoirs, particularly their ability to support chronic maintenance of the bacteria in their kidneys [14]; and indeed, only a limited number of species definitively fulfill this criteria. Rodents are considered as the main reservoirs, but other mammals, such as bats [15] and introduced [16,17] or endemic [18,19] small terrestrial mammals and cattle [20,21], have been identified as important reservoirs of *Leptospira*. The multiplication of pathogenic bacteria in animal reservoirs has been examined in experimental infections of mice under laboratory-controlled conditions [22]. Using bioluminescent *Leptospira*, authors have shown that a systemic infection associated with weight loss can occur within three days following intra-peritoneal injection. Thereafter, within a week, bacteria become rapidly invisible, while animals return to a body weight that is hardly distinguishable from that of control animals. Subsequently, a bioluminescent signal of *Leptospira* appears in the kidneys where bacteria actively divide leading to a glowing signal persisting for months and, at the same time, systemic infection has apparently vanished [22]. 

Hence, the fate of pathogenic *Leptospira* appears to be different in reservoir and incidental hosts, with a systemic infection followed by renal colonization in the former, contrasting with an absence of renal colonization in the latter. Discriminating between reservoir and incidental hosts may be not that clear cut, as it depends on different parameters [23]. Indeed, in experimental infections, the survival of infected animals and shedding of bacteria depend on the bacterial strains, the infecting bacterial dose, the considered vertebrate species, as well as the routes of infection [22,24,25,26]. For instance, though experimental infection of golden hamsters (*Mesocricetus auratus*) induces acute lethal infection in most animals, it may lead to chronic shedding in the few animals surviving the infection [27]. Of important note is that the colonization of renal tubules by leptospires, which is a characteristic of animal reservoirs, has a considerable immunological consequence: pathogenic *Leptospira* organized in biofilms in the lumen of renal tubules [28] remain hidden from the immune system and animals may not maintain in the long term the antibody responses elicited following acute infection. The duration of seropositivity is not clear and may not be for the life of the animal [29]. As a consequence, the immunological signature detected in an animal reservoir may not reflect the *Leptospira* species that are actually chronically shed by these animals. 

Beside its use in *Leptospira* classification, MAT is considered as the reference serodiagnostic test for leptospirosis in incidental hosts (humans and domestic animals), as it allows for detecting host antibodies that attest to current, recent, or past infections [6,30]. MAT has also been widely used for the investigation of animal reservoirs [31,32,33,34,35], but some studies indicate that MAT does not definitively identify the carrier status of a given animal species [36,37,38,39]. In the present work, we further support these latter observations and argue that the use of MAT may lead to incorrect conclusions regarding the role of investigated animal species as reservoirs and, hence, the importance of such animal species in the epidemiology of human forms of leptospirosis.

To support our purposes, we focused on mammal species known as pathogenic *Leptospira* reservoirs on southwestern Indian Ocean (SWIO) islands. This region is home to a wide diversity of mammals, including many endemic, as well as introduced rodents (family Muridae) and shrews (family Soricidae). The typing of *Leptospira* excreted by mammals in this region has demonstrated high levels of *Leptospira*-host specificity [18,19,40,41]. Indeed, the region is home to a large diversity of bats from seven different families that represent multiple colonizations of the region. Moreover, on Madagascar, endemic terrestrial mammals of the family Tenrecidae and family Nesomyidae (each representing separate colonization events and subsequent adaptive radiations), were shown to be the reservoirs of specific pathogenic *Leptospira*, providing interesting ecological circumstances to address the power of MAT for investigating leptospirosis epidemiology. 

Three animal species from the SWIO region, known to host distinct bacterial lineages/species of *Leptospira*, were included in the present investigation. Using published molecular and serological data together with original results, we demonstrate that *Leptospira* serological signatures are not necessarily connected to the *Leptospira* excreted by animal reservoirs. We demonstrate the shortfalls of only using MAT alone for the identification of *Leptospira* animal reservoirs and discuss the utility of MAT for clarifying leptospirosis epidemiology.

## 2. Materials and Methods

### 2.1. Ethical Considerations

All biological material screened in the present study was sampled in the context of a research program for which sampling methods, permit numbers, and IACUC acceptance have been presented elsewhere [40].

### 2.2. Animal Sampling, Leptospira Serological, and Molecular Data

The three mammal species from the SWIO analyzed in the study include the following: *Mormopterus acetabulosus* (Molossidae), an insectivorous bat species endemic to Mauritius; *Tenrec ecaudatus* (Tenrecidae), an omnivorous terrestrial mammal species endemic to Madagascar and introduced to several SWIO islands, including La Réunion and Mayotte; and an invasive rodent species, *Rattus rattus* (Muridae) sampled both on La Réunion and on Mayotte (Table 1 and Appendix A). All these species were previously investigated for *Leptospira* infection through molecular and/or serological methods by different research groups [19,21,35,40,42,43]. In addition, we produced serological data through MAT for *M. acetabulosus* samples. The same individual specimens were previously screened for *Leptospira* infection through molecular methods, and the sampling method for these animals is detailed in a published article [40]. The MAT was based on a panel of 18 *Leptospira* strains allowing for the screening of all serogroups recently reported on both human cases and animals from the Malagasy region. Strains are listed here as *Genus species* Serogroup/Serovar (type strain): *Leptospira biflexa* Semaranga/Patoc (Pato I [Paris]), *L. borgpetersenii* Ballum/Castellonis (Castellon 3), *L. borgpetersenii* Sejroe/Hardjobovis (Sponselee), *L. borgpetersenii* Sejroe/Sejroe (M 84), *L. borgpetersenii* Tarassovi/Tarassovi (Perepelicin), *L. interrogans* Australis/Australis (Ballico), *L. interrogans* Autumnalis/Autumnalis (Akiyami A), *L. interrogans* Bataviae/Bataviae (Van Tienen), *L. interrogans* Canicola/Canicola (Hond Utrecht IV), *L. interrogans* Hebdomadis/Hebdomadis (Hebdomadis), *L. interrogans* Icterohaemorrhagiae/Copenhageni (Wijnberg), *L. interrogans* Icterohaemorrhagiae/Icterohaemorrhagiae (Undetermined strain), *L. interrogans* Pomona/Pomona (Undetermined strain), *L. interrogans* Pyrogenes/Pyrogenes (Salinem), *L. kirschneri* Cynopteri/Cynopteri (3522C), *L. kirschneri* Grippotyphosa/Grippotyphosa (Moskva V), *L. kirschneri* Mini/Undetermined serovar (200803703) [35], and *L. noguchii* Panama/Panama (CZ 214K). Each serum was tested at dilutions ranging from 1:50 to 1:3200 and considered as positive when the MAT titer was ≥ 1:100.

## 3. Results

### 3.1. Bats

Serotyping of *Mormopterus acetabulosus* samples through MAT indicates that 20.0% (6/30) of specimens were seropositive, with sera agglutinating the Panama and Pyrogenes serogroups (Table 2 and Appendix A). Using nucleic acids extracted from the kidneys of the same individual specimens, Dietrich et al. [40] reported that 73.3% (22/30) of the animals tested positive through real-time polymerase chain reaction (RT-PCR), stressing the poor agreement between MAT and RT-PCR (Kappa test = 0.17). More specifically, the six MAT-positive bats were also positive by RT-PCR while, most importantly, 66.6% of the remaining MAT-negative bats (16/24) tested positive by RT-PCR. The sequencing of RT-PCR-positives specimens (also positive in MAT) confirmed that *M. acetabulosus* harbors a *Leptospira* bacterial sequence falling within the *Mormopterus*-borne *Leptospira* monophyletic clade embedded in *L. borgpetersenii* [40] (Table 2). 

### 3.2. Rats

Introduced populations of *Rattus rattus* are present on both Mayotte and La Réunion, but molecular and serological screenings of *Leptospira* highlight striking differences between these two islands (Table 2). On Mayotte, animal antibodies identified three previously reported serogroups, namely Mini, Pyrogenes, and Grippotyphosa, whereas on La Réunion, the main detected serogroups correspond to Icterohaemorrhagiae, Canicola, Sejroe, Mini, and Cynopteri [35,42]. Molecular investigations of *R. rattus* on both islands revealed sharp inter-island differences, with La Réunion animals harboring strictly *L. interrogans* [21], while on Mayotte, this rodent may harbor any of four distinct *Leptospira* species (*L. interrogans, L. borgpetersenii, L. kirschneri* and *L. mayottensis*) [19,42]. 

On La Réunion, a study investigated a human leptospirosis outbreak after a triathlon event. In summary, 10 *R. rattus* individuals were trapped a few weeks at the site the sporting event took place, and 5 out of the 10 sampled rats tested positive by PCR using kidney samples. The sequencing of the positive samples revealed only *L. interrogans* [21,43]; however, two of the PCR-positive specimens were seropositive through MAT, whereas the remaining PCR-positive rats were seronegative. 

### 3.3. Tenrecs

On La Réunion, antibodies detecting three serogroups in *Tenrec ecaudatus*: Icterohaemorrhagiae are usually detected at high titers, while Canicola and Bataviae serogroups are agglutinated at low titers [34,42] (Table 2). Notwithstanding these serological results, *T. ecaudatus* is not considered as a *Leptospira* reservoir on that island since renal carriage could not be demonstrated by two independent studies [21,42] (Table 2). This situation contrasts with that on Mayotte, where *T. ecaudatus* was identified as the reservoir of *L. mayottensis*, a pathogenic species commonly associated with human leptospirosis on that island [19] (Table 2). 

## 4. Discussion

The Microscopic Agglutination Test (MAT) has been, and still is, considered the gold standard for leptospirosis diagnosis in humans. In mammals, a meta-analysis has calculated the mean prevalence of infection using the data published in 300 publications including eight different taxonomic orders [33]. MAT and PCR were given an equivalent weight in that analysis, and the nature of the screened samples, i.e., blood (for MAT and PCR) or kidney/urine (for PCR only) was not taken into consideration. Hence, the discrimination between acute and previous infections in incidental hosts and chronic kidney carriage in true animal reservoirs was not made in that study, while this distinction has been made other studies. In the bat samples screened in the present study, we demonstrate a poor agreement between data generated using serological and molecular analyses. Similar findings were also reported on an Australian fruit bat species, *Pteropus alecto* (family Pteropodidae) [44], which indicated poor agreement between PCR (kidney detection) and serological data; these results underline that a carrier status for this bat species could not be shown using serology. In Brazil, studies have reported limited agreement or an absence of agreement between PCR (urine detection) and serological results in livestock animals or asymptomatic dogs [39,45]. Lastly, a comprehensive study recently reported the visualization, isolation, and genetic characterization of *L. interrogans* from a kidney sample obtained from a seronegative crested porcupine (*Hystrix cristata*) in Italy [46].

The bat species investigated herein belongs to the genus *Mormopterus*, which includes two other species in the SWIO region, *M. francoismoutoui* and *M. jugularis*, endemic to La Réunion and Madagascar, respectively. Recently, research conducted on these three molossid bats has shown that they shelter pathogenic *Leptospira* clustering into a single monophyletic *L. borgpetersenii* clade [40,41]. Interestingly, the screening of *M. acetabulosus* specimens through MAT reveals that sera agglutinate two distinct serogroups, i.e., Panama, and Pyrogenes. Although there is poor congruence between serogroups and genomospecies, members of Panama serogroup can be found in two species, *L. noguchii* and *L. inadai* [6], but not in *L. borgpetersenii*. This suggests that the Panama serogroup signature most likely results from independent systemic infections, which have dissipated without leading to renal colonization.

Although the molecular and serological analyses from *Rattus* and *Tenrec* were not all obtained from the same individuals, the results presented herein do not support a clear agreement between molecular and serological data. Interestingly, the investigation of these two mammal genera on Mayotte and La Réunion highlight the importance of independently evaluating the reservoir status of a given mammal species in different settings. Indeed, *Rattus* is reservoir of different *Leptospira* species on the two islands. The investigation of *Tenrec* is even more compelling. While on Mayotte, *T. ecaudatus* is the carrier of the recently described *L. mayottensis*, investigations on La Réunion showed that up to 81% of individual *Tenrec* were seropositive (mostly reacting against Icterohaemorragiae), but not a single individual showed evidence for chronic kidney infection [21,34,42]. On La Réunion, *T. ecaudatus* is therefore not a reservoir of *Leptospira* and the Icterohaemorragiae serogroup seropositivity revealed through MAT should be considered as a mere signature of past infections of these animals exposed to *Leptospira* present in their environment. 

In our tested samples, some individuals were positive through PCR but negative through MAT. These discrepancies may result from past infections and subsequent kidney colonization followed by antibody titer decay and eventual seronegativation, as previously reported in animal reservoirs [29,46] and incidental hosts [47,48,49]. We propose that conflicting results of known reservoirs, such as bats testing positive through MAT but negative through PCR using kidney tissues and/or urine, are best explained by animals that experienced past infections and for which *Leptospira* did not colonize the kidneys. This absence of colonization might be related to a low infecting dose and/or from an infecting bacterial genomospecies that is unable to establish persistent renal colonization in a specific vertebrate species. This hypothesis assumes that some putative host-*Leptospira* molecular determinants may be required for renal colonization and the assumption is supported by the fact that *Leptospira* from bats and tenrecs were not able to lead to chronic infection in rats [50]. In any case, investigation of the physiological and ecological conditions that determine renal tubule colonization and maintenance of the reservoir status are needed. 

Finally, the biological setting of the SWIO islands brings further evidence of problems using MAT for the identification of *Leptospira* reservoirs. Several studies have used MAT on samples of wild animals to address their role in the epidemiology of leptospirosis. However, even though it is clearly important to address the diversity and intensity of *Leptospira* exposure in an environmental setting, we emphasize that MAT data from investigated animals cannot provide a robust conclusion regarding their role as a reservoir. Such investigations, carried out in the One Health framework, require bacterial genotyping using kidney or urine samples so that bacteria excreted by mammal reservoirs can be compared to those identified in acute human cases. Only such comprehensive approaches can provide solid conclusions regarding the identification of the main reservoir(s) within an ecosystem and the actual risk for humans [51]. Such information is key to conceiving adapted preventive interventions aiming at reducing human contamination. 

## 5. Conclusions

The data presented and discussed herein strongly support that a thorough investigation of leptospirosis following the One Health framework requires (i) PCR screening of urine or kidney tissues from putative animal reservoirs, (ii) isolation of *Leptospira* from identified reservoirs, and (iii) the inclusion of these isolates in a MAT panel used to screen human sera collected from individuals with activities in the investigated environment. Such analyses allow the identification of animal reservoirs in a specific environmental setting, but also highlight those bacterial species/lineages of major medical concern. 

## Figures and Tables

**Table 1 tropicalmed-06-00205-t001:** Animal species used in the present study and the associated publications for *Leptospira* investigations.

Animal Species	Islands	Serological Data	Molecular Data
*Mormopterus* *acetabulosus*	Mauritius	Present study	[40]
*Tenrec ecaudatus*	Mayotte	NA	[19]
	La Réunion	[42]	[21,42]
*Rattus rattus*	Mayotte	[35]	[19,35]
	La Réunion	[42]	[21,42]

**Table 2 tropicalmed-06-00205-t002:** Comparison of serological and molecular *Leptospira* data obtained from the investigated animal species.

		Serological Data (MAT)		Molecular Data (RT PCR)	
Animal Species	Islands	Positive Animals (%)	Detected Serogroups (Titer)	Positive Animals (%)	*Leptospira* spp.
*Mormopterus acetabulosus*	Mauritius	20.0% (6/30) *	Panama (1:100–400) (*n* = 5) * Pyrogenes (1:200) (*n* = 1) *	73.3% (22/30) [40]	*Lb* (*n* = 8) [40]
*Tenrec ecaudatus*	Mayotte	*NA*	*NA*	27.0% (10/37) [19]	*Lm* (*n* = 8) [19]
	La Réunion	13.2% (5/38) [42]	Icterohaemorrhagiae (1:200–800) (*n* = 3) Canicola (1:100) (*n* = 1) Bataviae (1:100) (*n* = 1) [42]	0.0% (0/38) [42] 0.0% (0/25) [21]	-
*Rattus rattus*	Mayotte	11.2% (14/125) [35]	Mini (1:100–400) (*n* = 7) Pyrogenes (1:200) (*n* = 1) Grippotyphosa (1:100–1600) (*n* = 3) Co-agglutinations (*n* = 3) [35]	29.8% (42/121) [35] 15.9% (46/289) [19]	*Lb* (*n* = 9), *Li* (*n* = 7), *Lk* (*n* = 2), *Lm* (*n* = 2) [35] *Lb* (*n* = 13), *Li* (*n* = 3), *Lk* (*n* = 5) [19]
	La Réunion	78.8% (52/66) [42]	Icterohaemorrhagiae (1:100–3200) (*n* = 22) Canicola (1:100–400) (*n* = 7) Sejroe (1:100–1:200) (*n* = 2) Mini (1:100) (*n* = 1) Cynopteri (1:3200) (*n* = 1) Co-agglutinations (*n* = 19) [42]	65.8% (50/76) [42] 38.5% (214/562) [21]	*NA* *Li* (*n* = 201) [21]

*Lb*: *L. borgpetersenii*, *Li*: *L. interrogans*, *Lk: L. kirschneri*, *Lm*: *L. mayottensis.* MAT: Microscopic Agglutination Test, RT PCR: real-time polymerase chain reaction. *NA*: Not available. *: Original data.

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
