# Peer review of "Tracking Animal Reservoirs of Pathogenic *Leptospira*: The Right Test for the Right Claim"

_tropicalmed, 2021, doi:10.3390/tropicalmed6040205_

Round 1

Reviewer 1 Report

The manuscript proposed by Gomard and colleagues aimed to find the right test fot the leptospirosis diagnosis in wild animals.

The research is well conduct and the manuscript is well written although there are some typos need to be revised.

There are some point to be revised:

  • The leptospira species must be written in italic
  • Please report the methods used to sample live animals.
  • the comparison between your original data and just reported infection rate should be clarified.
  • Table 2 is very confusing, please revise it
  • line 226-233: please note that this seems to be common in wild animals, especially in animals that do not live in urbanized areas. See Coppola 2020 (https://pubmed.ncbi.nlm.nih.gov/32208192/) and Cilia 2020 (https://pubmed.ncbi.nlm.nih.gov/32558332/) reporting these differences.

Author Response

Dear reviewer,

Thank you for your comments and the different raised questions, all of which have been addressed, and which provide greater clarity to the manuscript. Please find below our responses to the points raised (in italics).

--------------------------------------------------------------------------------

The manuscript proposed by Gomard and colleagues aimed to find the right test fot the leptospirosis diagnosis in wild animals.

The research is well conduct and the manuscript is well written although there are some typos need to be revised.

There are some point to be revised:

  • The leptospira species must be written in italic

> Done, species names were italicized under Table 2 and in the reference section.

  • Please report the methods used to sample live animals.

>All included samples were obtained from previously published investigations and the requested details on trapping methods for these animals were detailed in the corresponding publications. However, for increased clarity, we modified one sentence in each of the Ethical considerations and methods sections, which now read (modifications are underlined):

Ethical considerations: “All biological material screened in the present study was sampled in the context of a research program for which sampling methods, permits numbers and IACUC acceptance have been presented elsewhere [40].”

Animal sampling, Leptospira serological, and molecular data:The same individual specimens were previously screened for Leptospira infection through molecular methods, the sampling method for these specimens is detailed in a previously published article [40]

  • the comparison between your original data and just reported infection rate should be clarified.

>We added a star on Table 2 indicating all original results and incuded the  publications corresponding to previously published data.

  • Table 2 is very confusing, please revise it

>There must have been a formatting problem, which now has been fixed and the table should now be clear.

  • line 226-233: please note that this seems to be common in wild animals, especially in animals that do not live in urbanized areas. See Coppola 2020 (https://pubmed.ncbi.nlm.nih.gov/32208192/) and Cilia 2020 (https://pubmed.ncbi.nlm.nih.gov/32558332/) reporting these differences.

>We added a reference to the work on Porcupine from Italy: “Lastly, a comprehensive study recently reported the visualization, isolation and genetic characterization of L. interrogans from a kidney sample obtained from a seronegative crested porcupine in Italy [47].”

Reviewer 2 Report

Summary

  Study from Gomard and colleagues recommend an advanced method in tracking animal reservoirs of pathogenic Leptospira. It is a well-written manuscript and will be interesting to those who are working on this field. I have little suggestions and questions for improvement.

  1. line 126-127, authors should describe the method of MAT in detail. This will allow the reader to more directly know the exact method, rather than going to another article.
  2. The differences of results examined by MAT and PCR may reflect different periods od infection. It is encouraging to combine multi methods to test animal reservoirs of Leptospira. Is this method suitable for interpreting the duration of leptospiral infection?
  3. Authors present a sequential methodology to investigate human leptospirosis and suggest to perform MAT using Leptospira isolates from reservoir animals occurring in the same environment. How much improvement could be achieved compared with previous method using some standard strains? Whether cross agglutination can mask the truth of infection?

Author Response

Dear Reviewer,

Thank you for your remarks which were all addressed.  Please note that our response to each point is in italics, and that line numbering refers to the revised version of the manuscript with track changes.

--------------------------------------------------------------------------------

Study from Gomard and colleagues recommend an advanced method in tracking animal reservoirs of pathogenic Leptospira. It is a well-written manuscript and will be interesting to those who are working on this field. I have little suggestions and questions for improvement.

  1. line 126-127, authors should describe the method of MAT in detail. This will allow the reader to more directly know the exact method, rather than going to another article.

>We detailed all the serogroups of the panel that was used to screen the sample (see lines 122-132). Please note that the original Table S1 was consequently deleted.

  1. The differences of results examined by MAT and PCR may reflect different periods od infection. It is encouraging to combine multi methods to test animal reservoirs of Leptospira. Is this method suitable for interpreting the duration of leptospiral infection?

>Yes, we thoroughly discuss the discrepancies between MAT and PCR in the discussion section, and we propose that they result from past infections, absence of kidney colonization and antibody decay following the acute phase. Such decay has been documented in nature for sea lions (see Lloyd-Smith et al, 2007). The period of acute infection averages one week in experimentally infected mice (Ratet et al., 2014), the titer decay for a given reservoir species could be measured through experimental infection and a longitudinal serosurvey, but is out of scope of the present work.

Authors present a sequential methodology to investigate human leptospirosis and suggest to perform MAT using Leptospira isolates from reservoir animals occurring in the same environment. How much improvement could be achieved compared with previous method using some standard strains? Whether cross agglutination can mask the truth of infection?

>There is indeed some level of cross reactivity between strains in MAT. However, some strains hardly cross react probably because of their genetic distance, such as L. interrogans Icterohaemoragiae and L. mayottensis Mini. In environments such as Madagascar, a MAT survey should provide insight into if people are mostly exposed to rat-borne L. interrogans or to tenrec-borne L. mayottensis, which is important information to decipher the epidemiology of the disease.

Round 2

Reviewer 1 Report

The authors' reply satisfy my comments. 

it remains only to add this requested reference (https://pubmed.ncbi.nlm.nih.gov/32208192/) among porcupine at line 213.

After this, I encourage the publication